# Detection of Destructive Processes and Assessment of Deformations in PP-Modified Concrete in an Air-Dry State and Exposed to Fire Temperatures Using the Acoustic Emission Method, Numerical Analysis and Digital Image Correlation

**DOI:** 10.3390/polym16081161

**Published:** 2024-04-20

**Authors:** Anna Adamczak-Bugno, Sebastian Lipiec, Peter Koteš, František Bahleda, Jakub Adamczak

**Affiliations:** 1Faculty of Civil Engineering and Architecture, Kielce University of Technology, Av. 1000-An. of Polish State 7, 25-314 Kielce, Poland; jakubadamczak123@gmail.com; 2Faculty of Mechatronics and Mechanical Engineering, Kielce University of Technology, Av. 1000-An. of Polish State 7, 25-314 Kielce, Poland; slipiec@tu.kielce.pl; 3Faculty of Civil Engineering, University of Žilina, Univerzitná 8215/1, 010-26 Žilina, Slovakia; peter.kotes@uniza.sk (P.K.); frantisek.bahleda@uniza.sk (F.B.)

**Keywords:** polypropylene concrete, fire conditions, acoustic emission method, numerical simulations, digital image correlation

## Abstract

This article presents the results of tests carried out to assess the condition of PP-modified concrete. The tests were carried out on samples previously stored at ambient temperature and exposed to temperatures corresponding to fire conditions—300 °C, 450 °C, and 600 °C. Axial compression tests of cube-shaped samples and three-point bending of beams were carried out. During strength tests, acoustic emission (AE) signals were recorded and the force and deformation were measured. Recorded AE events were clustered using the *k-means* algorithm. The analysis of the test results allowed for the identification of signals characteristic of the individual stages of the material destruction process. Differences in the methods of destruction of samples stored in ambient conditions and those exposed to fire temperatures were also indicated. While loading the samples, measurements were carried out using the digital image correlation (DIC) method, which enabled the determination of displacements. Based on the results of the laboratory tests, a numerical model was developed. The results obtained using different research methods (DIC and FEM) were compared. Tomographic examinations and observations of the microstructure of the tested materials were also carried out. The analyses carried out allowed for a reliable assessment of the possibility of using the acoustic emission method to detect destructive processes and assess the technical condition of PP-modified concrete. It was confirmed that the acoustic emission method, due to differences at low load levels, can be a useful technique for assessing the condition of PP-modified concrete after exposure to fire temperatures. So far, no research directions in a similar field have been identified.

## 1. Introduction

Concrete is widely used in construction; however, this material is susceptible to damage in extreme conditions. Typical examples of exceptional events are fire and high temperatures [1,2].

Factors that have a major impact on the degree of concrete damage at high temperatures include the rate of temperature increase, the maximum heat level, exposure time, the rate of cooling from the maximum temperature, the conditions prevailing after cooling, and the level of load transferred [3,4,5]. Despite the large impact of the factors mentioned, in reality the surrounding conditions may be difficult to predict, or more precisely, difficult to control [2,6,7].

High temperature adversely affects the properties of most cement composites. In fire conditions, concrete structures with a high moisture content and those made of high-strength concrete (HSC) tend to experience thermal spalling. This phenomenon mainly concerns concrete with a low water–cement ratio (*w*/*c*), high compressive strength, and a compact cement matrix [8,9,10].

The detachment of large fragments of concrete from the structure reduces the strength of the structure. As a result, this may lead to its damage. The phenomenon of “spalling” also poses a significant threat to rescuers conducting rescue operations [11].

The most effective methods of protecting structures against high temperatures and thermal explosive spalling include

use of a thermal barrier (fireproof insulation—surface protection with boards or a layer of shotcrete);using the addition of polypropylene to the concrete mix;adding an air-entraining agent to the concrete mix, using an aggregate with low thermal expansion [10,11].

There have been many scientific and research works aimed at, among other things, studying the positive impact of the addition of polypropylene (PP) fibers and particles on reducing the spalling phenomenon. Thermal explosive spalling of concrete was the subject of, among other subjects, the research covered by the UPTUN project [1,2,6,7].

In order to reduce the spalling of concrete in fire conditions, PP fibers or particles are added to the concrete mixture, generally in an amount from 1 kg/m^3^ to 2 kg/m^3^. This amount corresponds to 0.4–0.8% of the volume of the cement slurry, assuming that the cement slurry occupies 30% of the volume of concrete. In order to obtain good dispersion during mixing, the fibers or particles are coated with an agent that reduces their hydrophobic properties. These may be agents such as fatty acid esters, glycerides, fatty acid amides, or cationic surfactants [2,6,7,8].

The most popular varieties of polypropylene plastics soften and begin to melt at a temperature of approximately 160 °C. Under such conditions, the volume of the additive decreases. In the spaces left by the filler, channels are formed through which water vapor escapes under high pressure. Thanks to this, internal stresses do not reach a critical point and explosive spalling of the concrete does not occur [10,11,12,13].

The technological procedure described above allows counteracting the phenomenon of explosive spalling, primarily by reducing the maximum values of water vapor pressure by increasing the permeability of concrete. The increase in permeability occurs due to the melting of fibers or particles and their pyrolysis [12,13].

As a result of fire impact on concrete structures, various damages may occur, the scope and intensity of which depend on the detailed construction and material solutions used in the facility, the type and size of the impacts, and the actual course of temperature changes during the fire. In order to properly assess the technical condition and safety level of a structure after a fire and decide on the scope of possible repair activities, it is necessary to analyze the impact of the identified damage on the load-bearing capacity of the structure, based on reduced material parameters determined taking into account the unfavorable effects of high fire temperature. Since the duration of an actual fire is limited and concrete structures are usually not completely destroyed during a fire, determining the technical condition and conducting an analysis of the safety level of the structure after the fire is necessary to make the right decision regarding the strategy for repairing/strengthening the structure as an alternative to demolition [14,15,16,17].

Diagnostic techniques used to assess the condition of concrete in structures damaged by high temperatures can be divided into tests performed in situ and tests conducted in laboratory conditions on specimens taken from the structure. In the case of local assessment of concrete quality, in situ tests are most often performed using non-destructive and semi-destructive methods, which are commonly used to control the properties of concrete in structures [18,19,20,21].

Among the methods for assessing fire damage to concrete, techniques that allow for a comprehensive assessment of a structural element occupy a special place. These methods use physical phenomena related to the propagation of electromagnetic waves (GPR) or phenomena related to the propagation of surface waves—analysis of surface waves (multichannel analysis of surface waves). Both techniques were taken from geotechnics and allow obtaining isoline maps of material properties for the analyzed elements; however, their use in assessing fire damage is still pioneering and requires further research and analysis [17,18,19,20,21,22].

Laboratory tests of concrete, aimed at assessing degradation and estimating the depth of damage, require collecting material from elements damaged during a fire, most often by drilling. Tests carried out on drilling holes concern the determination of the mechanical properties of concrete. Their purpose is also to determine the condition of the material’s interior. For this purpose, the following methods are used: X-ray, scanning microscopy, DTA and TGA tests, thermoluminescence, colorimetry, and chemical or petrographic analysis, as well as evaluation methods based on porosity measurements using mercury porosimetry or by assessing the density of micro-scratches [17,18,19,20,21,22,23,24,25,26].

The acoustic emission (AE) method has been used for a long time to monitor the condition of many engineering facilities. Acoustic emission is defined as transient elastic waves arising as a result of local internal micro-displacements in the material. The AE method is already a common NDT method used mainly for testing stationary equipment—tanks, pressure vessels, reactors, pipelines, bridges, etc. The AE method has been fully accepted and standardized for some applications. AE provides a large set of information about the response of materials to applied loads. It is effective for detecting and identifying increasing material defects. Thanks to its sensitivity, it can detect processes such as the formation and development of microcracks, the movement of dislocation groups, cracking, sliding, and separation of sediments [24,25,26,27].

An advantage of the AE method is the ability to examine the entire structure/element, locate any damage, and assess its threat to the integrity of the structure. AE application also brings significant economic benefits [27,28].

Acoustic emission is a technique capable of detecting, classifying, and locating damage in concrete by capturing and analyzing acoustic emission signals generated during compression, splitting, or other mechanical processes. The AE technique has been used to investigate damage to structural materials [29,30]. Suzuki et al. discovered the relationship between AE energy rate and physical properties and made a correlation between the parameters, and a qualitative assessment of concrete damage was made based on AE signals [30]. Suzuki et al. also used the AE technique and damage mechanisms to quantify the damage of concrete structures [27]. Wang et al. studied the acoustic emission characteristics and the failure process of mortars with polypropylene fibers and concrete reinforced with steel fibers at different strain rates [28]. Their study has important value for understanding the damage development characteristics of fiber-reinforced concrete. By comparing the AE performance characteristics of cementitious materials under bending and shear loading, Farhidzadeh et al. realized the fracture mode identification for tensile and shear fracture [31]. Kawasaki et al. used continuous acoustic emission monitoring to study the corrosion damage of reinforced concrete beams under wet–dry cycles and determined the occurrence of corrosion and the nucleation of corrosion cracks [32]. The results showed that the AE technique has great prospects for the quantitative assessment of concrete in the air-dry state and exposed to fire temperatures. It was concluded that the AE signal is sensitive to subtle damages inside the materials and is suitable for damage analysis of composite materials [33,34,35,36,37,38].

This article identified destructive processes in PP-modified concrete during compression and bending tests. For this purpose, a multi-criteria analysis of AE signals was used. Differences in the range of work under load were indicated for samples stored at room temperature and exposed to fire temperature. It was established that high temperatures lead not only to a decrease in the mechanical parameters of the material, but also to a change in the way it functions under load. It was confirmed that the acoustic emission method, due to differences at low load levels, can be a useful technique for assessing the condition of PP-modified concrete after exposure to fire temperatures. So far, no research directions in a similar field have been identified.

## 2. Materials and Methods

### 2.1. Materials

The tests were aimed at checking the behavior of the material under compressive and bending loads. Specimens for axial compression tests took the form of concrete cubes with dimensions of 150 × 150 × 150 mm. The structure of the material was concrete with the addition of dispersed PP particles. Specimens for three-point bending tests took the form of concrete beams with dimensions of 50 × 50 × 300 mm. The structure of the material was concrete with the addition of dispersed PP particles.

The research material was obtained from an entity offering the sale of modified concrete for industrial applications. The manufacturer of the research material was KAPIT s.r.o. (Žilina, Slovak Republic). Information regarding the exact composition of the material was not given, because the recipe is protected by patent law.

Four series of three specimens were tested in both compression and bending tests. Specimens from the first group (*C*_1_ for compression and *B*_1_ for bending) were stored at ambient room temperature (±20 °C). Specimens from the second test series (*C*_2_ for compression and *B*_2_ for bending) were fired in a laboratory furnace. The processing temperature was 300 °C. The heating time from the moment the set temperature was reached was 3 h. Specimens from the third test series (*C*_3_ for compression and *B*_3_ for bending) were fired in a laboratory furnace at a temperature of 450 °C. The heating time from the moment the set temperature was reached was 3 h. Specimens from the fourth test series (*C*_4_ for compression and *B*_4_ for bending) were fired in a laboratory furnace at a temperature of 600 °C. The heating time from the moment the set temperature was reached was 3 h.

The firing time used for the samples made it possible to obtain the same temperature throughout the entire volume of the conditioned elements. Samples cooled in air at room temperature were tested.

### 2.2. Methods

#### 2.2.1. Acoustic Emission Method

As mentioned, acoustic emission (AE) is a non-destructive method for tracking the growth of cracks in the structure of a material subjected to deformation or internal stress. The method is based on monitoring the mechanical energy released in the form of elastic waves when the structure of a deformed material cracks. This energy passes through the material in the form of ultrasonic and audible sound waves and is measured on the surface of the object using a piezoelectric transducer (microphone). By digitally recording and processing individual AE events in real time, the method allows, not only constant monitoring of crack propagation, but also prediction of the macro damage that precedes a noticeable increase in the level of AE activity.

NOESIS 12.0 software is useful for classifying AE signals and is based primarily on the pattern recognition method. Programs from this group enable analyses with automated division into classes (unsupervised)—USPR and in the self-learning variant, in which division into classes is carried out using reference signals (supervised)—SPR.

The NOESIS program uses various grouping methods, but the instructions do not provide guidelines for choosing one of them. In the analyses discussed, the *Fuzzy K-means* algorithm was used.

Clustering is one of the unsupervised learning techniques. The clustering problem involves dividing data into groups (clusters) that contain elements that are similar to each other. First, an initial division of the data is performed. Then, the obtained division is modified in such a way that some elements are transferred to other groups, so as to obtain the minimum variance within each group. The aim is to ensure that the similarity of elements within each group is as high as possible, but at the same time ensuring the greatest possible difference between groups [38].

Data grouping methods can be divided into two categories: hierarchical methods and non-hierarchical methods. The *k-means* method is one of the non-hierarchical methods. The main difference between hierarchical and non-hierarchical methods is that in the case of non-hierarchical methods it is necessary to specify the number of groups in advance. Data grouping is widely used in many different areas (fields), such as biology, medicine, economics, and image processing [38].

An appropriate evaluation function is responsible for the homogeneity of the elements in each group. There are many different scoring functions that can be used to measure the quality of clustering. The most frequently used function is called the sum of squared error (SSE), which can be defined as the sum of the squared distances between the elements in the dataset and the centroids to which these elements are assigned. The most commonly used distance function is Euclidean distance [33,38].

A popular and widely used algorithm is the *k-means* algorithm. For a given dataset and initial solutions (centroids), in subsequent iterations, the algorithm generates better and better solutions in which the SSE value is smaller and smaller. The iteration of the algorithm can be divided into two main phases: the assignment phase and the centroid calculation phase. In the assignment phase, each element of the dataset is assigned to the nearest centroid. Then, in the centroid calculation phase, they are recalculated as the average value of the assigned training vectors. Individual phases of the algorithm are repeated until the appropriate stopping criterion is reached. The most common convergence criterion is a predetermined number of iterations or a step at which the affiliation of dataset elements to centroids has not changed [35,36].

#### 2.2.2. Digital Image Correlation DIC

The development of non-invasive methods for evaluating the strain process of loaded components/specimens makes it possible to increase the amount and quality of information obtained from laboratory tests. One of the many methods used to measure material deformation is digital image correlation. The principle of using DIC in testing is based on analyzing the image of the strain of the specimen surface, over the entire load range. The area of the specimen to be recorded during the test should be properly prepared by applying the appropriate quality of points. The texture should be applied to the object before the measurements start. The method and technique of applying the texture to the surface of the specimen requires the experience of the person carrying out the tests using image registration. One of the recorded images is defined as the reference image for all subsequent analyses. Software using DIC analyzes changes in the position of each point registered on the reference image, over the entire load range [39,40,41,42,43,44,45]. The DIC analysis results in displacement values from which, in the next step, strains are determined from simple relationships. In the research considered in this article, a so-called videoexensometer and analysis using GOM Correlate software ver. 2023.3.0.969 were used [46]. Strain levels determined in the material using digital image capturing (DIC) methods for the loaded element become a natural way to verify obtained strain results from other sources—e.g., numerically determined [38,39,47].

#### 2.2.3. Numerical Simulations

In order to determine the values of stresses and deformations in the modified concrete adopted for the analysis, load simulations of compression cubes and three-point-bending beams were carried out using the finite element method (FEM). Numerical models were developed and calculations were carried out in Abaqus ver. 2017 [48]. Numerical models were developed: three-dimensional models of the compression cubes, and two-dimensional models of the three-point-bending specimens. Dimensionally, the specimens were identical to those used in laboratory tests. The correctness of numerical results obtained is primarily determined from two aspects: the use of a suitably prepared material compound, and the discretization of the numerical model, which involves careful development of a high-quality finite element (FE) grid. Due to the nature of the material under consideration—concrete with polypropylene particles—the option available in Abaqus—concrete damage plasticity (CDP)—was used to define the material model [38,47,49,50,51,52,53,54,55]. Information on the material model used is presented in Table 1.

Eight-node (type C3D8R for the compression specimen) and four-node (type CPS4 for the flexural specimen) finite elements were used in the discretization of the numerical models. The model of the compression specimen contained 8000 nodes, while the model of the three-point bending specimen contained 1820 nodes. The quality of the finite element mesh was verified using the built-in option in Abaqus. No low-quality elements were shown. When defining the boundary conditions in the compressed cube, the possibility of displacement of the bottom wall of the model was blocked, while displacement of the top wall was enforced (Figure 1a). The ability to move the two rollers, located at the bottom of the numerical specimen model, was disabled. The deflection of the numerical beam model was forced by applying the displacement of the upper roll (Figure 1b). In the case of the numerical models of both analyzed specimens, the displacement value declared in the numerical calculation program corresponded to the displacement value at the maximum force loading the element, recorded during the laboratory tests.

#### 2.2.4. CT Computed Tomography

CT computed tomography is a type of X-ray spectroscopy, a diagnostic method that allows obtaining layered images of the examined object. It uses a composite of object projections made from different directions to create cross-sectional (2D) and spatial (3D) images [56,57,58].

Creating a tomographic image involves measuring the absorption of radiation passing through an object. The volume of the object is divided into small cells, called voxels, in which the linear radiation absorption coefficient is the same. The reconstructed cross-sectional image is a quantitative map of the linear radiation absorption coefficient in the voxels included in the scanned layer. The distribution of radiation absorption coefficients is calculated by a computer, which is why the method is called computed tomography [59,60].

Modern tomographic methods used in relation to building materials make it possible to obtain information regarding the diagnosis of the destruction process and the relationship between microstructure and functional properties. The development of tomographic methods is mainly aimed at detecting various types of defects in the tested material [61,62].

The research discussed in this manuscript used the Nikon XT H 225 ST (Nikon, Leuven, Belgium) computed tomography system, which allows the examination of details weighing up to 50 kg. This advanced computed tomography system equipped with a metrology package allows you to conduct research in accordance with the VDI 2630 standard [63] and obtain the same or more accurate measurements than coordinate measuring machines.

The Nikon XT H 225 ST system consists of the following components:a high-energy X-ray tube generating radiation up to 225 kV at a power of 450 W;Perkin Elmer 1620 digital detector with a pixel size of 200 μm;five-axis object manipulator with an integrated rotary table;granite manipulator base with an inspection table;computer with dedicated software.

The tested specimens were exposed to ionizing radiation, as a result of which a series of 2D images were obtained at various angles of rotation; then, the images were reconstructed and a 3D model was obtained.

The test parameters for the tested specimens were determined experimentally and were assumed:X-ray tube exposure parameters—215 kV, 355 μA;distance between the lamp and the detector—600 mm;exposure time—354 ms;number of projections—4476;image resolution—31 μm;geometric magnification—5×;voxel size—27 μm.

Data processing was carried out in the VG Studio Max program.

## 3. Results

### 3.1. Results of the Laboratory Tests

Two types of tests were to be carried out in the laboratory testing program: a compression test and a three-point bending test. For the compression test, the specimens were 150 × 150 × 150 cubes, while the three-point bending test used beams measuring 300 × 50 × 50. The test specimens were made of concrete together with PP-particle reinforcement, after heat exposure of the analyzed material (as introduced in the first part of the article). Laboratory tests were carried out on a Zwick testing machine. The following signals were recorded throughout the tests: test time, force applied to the specimen, and displacement of the machine traverse. In addition, an image of the deforming specimen was recorded during running time, for subsequent analysis using DIC. Graphs of specimen loading force as a function of traverse displacement are shown in Figure 2a for compression tests and Figure 2b for three-point bending tests.

For the compression case, the highest level of force loading of the specimen was recorded for the material designated as *C*_1_—without burning the concrete under fire conditions. This was a value of over 500 kN. As temperature was applied (increased) to the concrete material through annealing, a gradual decrease in the maximum force recorded during the compression tests was obtained. A significant decrease in the loading force on the specimen occurred when the annealing temperature of the concrete was increased above 300 °C—material *C*_2_. With the thermal effect on the concrete material, an increase in the recorded displacement of the traverse was obtained, i.e., the level of nominal strain increased (Figure 2a).

In the case of the three-point bending specimens, significantly higher values for the maximum specimen loading force were obtained for concrete material without baking (*B*_1_) and concrete subjected to baking at 300 °C (*B*_2_). A decrease in the specimen loading force to a value of 90–120 N was recorded for concrete annealed at the two highest temperatures analyzed: 450 °C (*B*_3_) and 600 °C (*B*_4_). As the firing temperature of the concrete containing PP particles increased, the material was able to sustain a higher value of traverse displacement (above 3 mm for materials designated *B*_3_ and *B*_4_) (Figure 2b).

The appearance of the specimens after the compression test and the three-point bending test is shown in Figure 3a–d. After the compression test, the specimen made of concrete without furnace baking showed no visible loss of material (Figure 3a). The same was true for the concrete annealed at 300 °C (Figure 3b), but in this case cracks and areas of partial material disintegration were already visible on the surface of the specimen. For concrete with PP particles annealed above 300 °C (material designations *C*_3_ and *C*_4_), the specimens were significantly crushed after the compression test, with material loss and with the polypropylene particles melted during annealing visible (Figure 3c,d).

For the concrete material designated *B*_1_ and *B*_2_, the specimens after the three-point bending test did not show any visible significant cracks or losses (splits) in the material (Figure 3a,b). From concrete annealing temperatures above 300 °C after the bending test, the occurrence of concrete cracks could be observed on the specimens, as visible in Figure 3c,d.

The information obtained from the laboratory compression and three-point bending tests on concrete specimens reinforced with PP particles, taking into account observations of the loading behavior of the specimens, formed the basis for developing numerical models and carrying out finite element method (FEM) simulations.

### 3.2. Results of Measurements and Analyses Using the Acoustic Emission Method

In order to create a database of reference signals for the destructive processes occurring in the material of concrete specimens modified with PP particles, 14 parameters of EA signals were used:Duration;Rise time;Decay time;RMS;Counts;Counts to peak;Amplitude;Energy;Average frequency;Reverberation frequency;Initiation frequency;Absolute energy;Signal strength;Average signal level (ASL).

The degree of matching of signals to individual classes was assessed based on the value of the R^2^ coefficient. The value of the indicator in each case was not less than 0.9.

#### 3.2.1. Results of Acoustic Emission Analyses for Compressed PP-Modified Concrete Cubes

The development of the material crack sequence during compression was divided into seven stages, taking into account the level of micro- and macro-cracks. Characteristic processes were assigned to classes of acoustic emission signals isolated using the *k-means* algorithm:elastic deformations in the linear range—class 1 (red);crack initiation on a micro scale—class 2 (blue);development of cracks on a micro scale—class 3 (black);coalescence of cracks on a micro scale—class 4 (pink);crack initiation on a macro scale—class 5 (green);emergency condition—class 6 (orange).

Table 2 shows the ranges of some acoustic emission parameters for each grouping class.

Analyzing the graph of the energy parameter of acoustic emission signals of individual classes in time for an exemplary specimen from the *C*_1_ series (Figure 4), it can be seen that in the time course from the beginning of the load application there were signals corresponding to the elastic work of the material. Shortly after the beginning of the study, signals related to the initiation of cracks in the microstructure and their development appeared. The increase in force led to a gradual coalescence of deformations. The occurrence of damage on a macro scale led to rapid destruction of the element. Subsequent processes were characterized by increasing energy.

Analyzing the graph of the energy parameter of acoustic emission signals of individual classes over time for an exemplary specimen from the *C*_2_ series (Figure 5), it can be seen that in the time course from the beginning of the load application there were signals corresponding to the elastic work of the material. Shortly after the beginning of the study, signals related to the initiation of cracks in the microstructure and their development appeared. The increase in force led to a gradual coalescence of deformations. The occurrence of damage on a macro scale led to rapid destruction of the element. It was noticed that for the samples from the discussed series there was a break in the development and coalescence of cracks on a micro scale. Subsequent processes were characterized by an increase in energy.

Analyzing the graph of the energy parameter of acoustic emission signals of individual classes over time for an exemplary specimen from the *C*_3_ series (Figure 6), it can be seen that signals of all classes appeared in the time course from the beginning of the test. The material destruction process proceeded at a faster pace compared to specimens in an air-dry state and fired at a temperature of 300 °C. Subsequent processes were characterized by signals of increasingly higher energy.

Analyzing the graph of the energy parameter of acoustic emission signals of individual classes over time for an exemplary specimen from the *C*_4_ series (Figure 7), it can be seen that signals of all classes appeared in the time course from the beginning of the test. The material destruction process proceeded at a faster pace compared to specimens in an air-dry state and fired at a temperature of 300 °C. Subsequent processes were characterized by signals of increasingly higher energy.

#### 3.2.2. Results of Acoustic Emission Analyses for Bending Beams Made of PP-Modified Concrete

In the case of the bending specimens, the best matching of the signals was obtained by dividing the events into four classes. The acoustic emission signals were attributed to the following processes:elastic work/micro crack initiation—class 1 (red);formation and propagation of cracks—class 2 (blue);development of cracks, crushing of concrete—class 3 (black);plastic deformation, material damage—class 4 (purple).

Table 3 shows the ranges of some acoustic emission parameters for individual grouping classes.

Analyzing the graph of the energy parameter of acoustic emission signals of individual classes over time for an example specimen from the *B*_1_ series (Figure 8), it can be seen that in the time course from the beginning of the load application there were signals corresponding to the elastic work of the material and the formation of microcracks. Shortly after the beginning of the study, signals related to the formation and propagation of cracks appeared. The increase in force led to the gradual development of deformations and crumbling of the concrete. The occurrence of damage on a macro scale quickly led to the destruction of the element. Subsequent processes were characterized by signals of increasingly higher energy. Signals associated with significant deformation and crushing of concrete appeared when the breaking force reached 50%.

Analyzing the graph of the energy parameter of acoustic emission signals of individual classes over time for an exemplary specimen from the *B*_2_ series (Figure 9), shortly after the beginning of the study, signals related to the formation and propagation of cracks appeared. The increase in force led to the gradual development of deformations and crumbling of the concrete. The occurrence of damage on a macro scale quickly led to the destruction of the element. Subsequent processes were characterized by signals of increasingly higher energy. Signals associated with significant deformation and crushing of concrete appeared when the breaking force reached 50%.

Analyzing the graph of the energy parameter of acoustic emission signals of individual classes over time for an example specimen from the *B*_3_ series (Figure 10), it can be seen that in the time course from the beginning of the load application there were signals corresponding to the elastic work of the material and the formation of microcracks. The total number of recorded signals compared to the *B*_1_ and *B*_2_ series specimens was much lower. The material destruction process proceeded at a faster pace compared to specimens in an air-dry state and fired at a temperature of 300 °C. Shortly after the beginning of the test, there were signals related to the formation and propagation of cracks, as well as evidence of the development of deformations and the crumbling of concrete. The occurrence of damage on a macro scale quickly led to the destruction of the element. Subsequent processes were characterized by signals of increasingly higher energy. Signals associated with significant deformation and crushing of concrete appeared after reaching 10% of the destructive force.

Analyzing the graph of the energy parameter of acoustic emission signals of individual classes in time for an example specimen from the *B*_4_ series (Figure 11), it can be seen that in the time course from the beginning of the load application there were signals corresponding to the elastic work of the material and the formation of microcracks. The total number of recorded signals compared to the *B*_1_ and *B*_2_ series specimens was much lower. The material destruction process proceeded at a faster pace compared to specimens in an air-dry state and fired at a temperature of 300 °C. Shortly after the beginning of the test, there were signals related to the formation and propagation of cracks, as well as evidence of the development of deformations and crumbling of concrete. The occurrence of damage on a macro scale quickly led to the destruction of the element. Subsequent processes were characterized by signals of increasingly higher energy. Signals associated with significant deformation and crushing of the concrete appeared after reaching 10% of the destructive force.

### 3.3. Results of the Numerical Analyses

As a result of the numerical simulations, the stresses and strains occurring in the numerical models of the specimen geometries analyzed were determined. The purpose of this was to determine the characteristic magnitudes of the mechanical fields accompanying concrete with PP reinforcing particles after exposure to high fire temperatures. In the case of stresses, the magnitudes of Mises effective stresses and stresses in the direction of displacement action in the numerical model (*σ*_22_) were analyzed. These are the stress levels characteristic of when the maximum force is reached in laboratory tests (Figure 2). Maps of the effective stress distributions in the modelled specimens for different annealing temperatures are shown in Figure 12. The characteristic stress magnitudes are summarized in Table 4.

Concrete with polypropylene particles stored at room temperature (+20 °C) had the highest levels of Mises effective stresses and the component acting in the direction of the applied load, for both compression and three-point bending tests (Table 4). The maximum value of effective stress was approximately 27.5 MPa in bending, and almost twice as much at approximately 47 MPa in compression. As higher and higher annealing temperatures were applied to the material of the analyzed concrete, a significant reduction in stresses was achieved, so that at an annealing temperature of 450 °C and above, the effective stress level was within 3 MPa, with *σ*_22_ stresses of less than 1 MPa for three-point bending. When observing the stress maps obtained from the numerical simulations (Figure 12a,c) for the bending test, it can be seen that for concrete stored at room temperature and baked at 300 °C, the maximum stress levels occurred in the center of the specimen, at its top edge, in close proximity to the contact of the roll loading the material. For the high annealing temperatures of concrete with PP particles (450 °C and 600 °C), the zone of maximum stresses moved away from the top surface of the specimen. This effect was indirectly visible with the image of the beam annealed at 600 °C, while a significant shift of the area of maximum stress values was recorded for the specimen annealed at 450 °C (Figure 12e,g). The differences in the stress distributions in the analyzed material were due to the influence of the melting of PP particles in the concrete, which could be seen in the laboratory test results (Figure 2).

The strains occurring in the material due to loading were also determined numerically. The analysis was based on the magnitude of the effective strain. This was in order to be able to verify the numerical results obtained with digital image correlation of the specimen during laboratory testing. Maps of strain distributions obtained from the DIC: for compression and three-point bending specimens are shown in Figure 13. The numerical values of strains determined numerically and those obtained from the analysis using DIC are summarized in Table 5.

In the case of the numerical analysis of the compressed cube, the lowest level of effective strain occurred for concrete with PP particles without baking, in the range of 23.5%. With the implementation of concrete annealing in the fire temperature range, the material showed a high level of strain, well above 40%. The maximum strain occurred for the specimen annealed at 300 °C and was over 57%, a gentle decrease in strain was recorded for the material annealed at 450 °C and 600 °C—within 50% (Table 5).

The lowest level of effective strain for the three-point bending mode that was determined by numerical simulations was recorded for the specimen without annealing under fire conditions (storage temperature +20 °C). The deformation was almost 13%. As the annealing temperature of the material increased, the effective strains increased, to almost 24% at temperatures: 300 °C and 600 °C. The highest strain value of 42% occurred at an annealing temperature of 450 °C. The area of the highest strain values was located in the central part of the specimen, at the point of contact between the material and the loading roller (Figure 13f,h). The increase in numerically determined strains when the maximum force was recorded on the experimental curve was identified with the phenomenon of the melting of PP particles, which were components of the concrete analyzed. This allowed for greater deformation of the concrete material as a result of loading. For each of the cases analyzed, a higher level of effective strain occurred for the case of concrete compression, relative to three-point bending.

An important aspect of the analyses presented was the possibility to verify the numerical results obtained in terms of strain. A second source for determining effective strain levels in the loaded material was analyses based on image registration of the DIC specimen. In the results of the analyses carried out using GOM software ver. 2023.3.0.969, it was possible to compare strain levels. In the case of the four analyzed variants of thermal action on concrete, the maximum difference between the numerical results and those determined by using the DIC was 4.11% for the treatment variant designated *C*_4_ and *B*_4_ (Table 5). An important aspect from the point of view of DIC analysis is the appropriate application of contrasting dots on the recorded specimen surface and the use of a camera of sufficiently high quality to record images of the deforming specimen.

### 3.4. Computed Tomography Results

Research using computed tomography involved the analysis of image sections in the XY plane and 3D models, including porosity measurement.

Figure 14 shows the distribution of 2D cracks along the loading direction (Z axis) in the XY plane (perpendicular to the loading direction). Inside the material, there were primary pores and microcracks (Figure 14a), which most often occurred at the edge of unhydrated cement, aggregate grains, and PP particles. The distribution of PP particles was non-uniform, and their size in cross-section/plan was comparable to the aggregate grains. Firing at a temperature of 300 °C resulted in the formation of new cracks, most often connecting the components of the cement matrix (Figure 14b). It is clearly visible that the PP particles have spread. Their size compared to the air-dry material was several times larger. In the analyzed image section, it can be seen that the plasticized filler has penetrated some of the cracks and pores that have formed. As the temperature increased, further development of primary cracks and the gradual formation of new damage throughout the material cross-section were noted. The cracks increased in scale and had complex shapes and modes (Figure 14c,d). 

During the observations, it was determined that the voids (pores, cracks) in the volume of individual specimens were as follows: 1.36%, 12.37%, 22.55%, 17.14%. The obtained values correlate with the previously discussed displacement results (Figure 15).

The results obtained using CT confirmed the authors’ assumptions as to the condition of the material after thermal conditioning and the influence of the structure on the emitted AE signals.

## 4. Discussion and Conclusions

Based on the analyses of the obtained research results regarding the destruction process of compressed cubes and bending beams made of modified PP concrete stored at ambient temperature and subjected to the influence of fire temperatures, performed using AE signal analyses, numerical analysis, DIC and computed tomography tests, certain generalizations were formulated.

Loading concrete modified with PP particles causes the occurrence of various mechanisms (processes) of material destruction, which include elastic deformations in the linear range, initiation of cracks on a micro scale, formation of cracks on a macro scale, crushing of concrete, and a state of destruction.Similar destruction processes were found for materials stored at ambient temperature and exposed to fire temperatures.The evolution of destructive processes in PP-modified concrete is associated with the emission of increasingly higher energy signals.The destruction of materials stored at ambient temperature and exposed to fire temperatures involves the emission of signals of similar energy.In the case of cubes and beams stored at ambient temperature and exposed to fire at 300 °C (series C1, C2, B1 and B2), the processes occurred sequentially until the elements had been destroyed. The course of signals of individual acoustic emission classes was close to linear.In the case of compression samples exposed to a fire temperature of 300 °C (series C2), a break in the development and coalescence of cracks at the micro scale was observed. This fact was referred to the possibility of local strengthening of the material as a result of increasing the degree of compactness of the material structure by the melted PP material.In the case of cubes and beams exposed to fire temperatures of 450 °C and 600 °C (series C3, C4, B3 and B4), a significantly smaller number of signals and a change in their distribution over time were observed.In the case of cubes and beams exposed to fire temperatures of 450 °C and 600 °C (series C3, C4, B3 and B4), all destructive processes were found to occur in stages at specific time intervals. This fact was related to the increase in the degree of porosity of the material due to high temperatures.As a result of numerical simulations, the highest levels of effective stresses and in the tensile direction (deflection) were determined for specimens stored at room temperatures. A significant decrease in stress values occurred with the concrete annealing temperature of 450 °C (from samples designated *C*_3_ and *B*_3_, respectively). The trend of the results was consistent with the recorded results of the laboratory tests. This relationship was present for both compression and three-point bending tests.The impact of fire temperature conditions on the concrete material increased the values of deformations determined in the elements based on finite element calculations. For the compression test, the highest levels of effective deformation occurred for specimens annealed at 300 °C (*C*_2_), while for the three-point bending test, it was for the material after annealing under temperature conditions of 450 °C (*B*_3_).The results of the numerical calculations were validated by comparing the obtained effective strain levels with the values determined using digital image correlation (DIC). The maximum difference was less than 5%.After analyzing the research results, the following conclusions were drawn:Fire temperatures of 300 °C, 450 °C, and 600 °C reduce the mechanical strength of PP-modified concrete under both compressive and bending loads.Compression and bending of PP-modified concrete causes the emission of acoustic signals characteristic of various destructive processes occurring in the material.Fire temperatures change the number and distribution of acoustic emission signals occurring during compression and bending of PP-modified concrete.Differences in the number and distribution of acoustic emission signals occur at low load and strain values.Analysis of acoustic emission signals according to the energy parameter over time allows drawing conclusions about the advancement of destructive processes taking place in the material structure and a preliminary assessment of the material’s technical condition.The acoustic emission method can be effective for monitoring the condition of PP-modified concrete elements or structures under load. This applies to both materials operated at ambient temperatures and those exposed to fire temperatures.The impact of high post-fire temperatures on concrete with PP particles affects the stress and effective strain distributions occurring in the material. Such a case occurs both for compression specimens and those subjected to three-point bending.An important aspect at the stage of validation of the numerical model is the possibility of comparing the determined values of deformations with those obtained as a result of implementation at the level of laboratory tests of digital image correlation. The correctness of the application of DIC analysis is conditioned by the experience of the person handling the preparation of the surface of the specimen for testing and the use of equipment for recording the image of the specimen with sufficient quality.The results of laboratory tests, registration and analysis of acoustic emission signals, numerical simulations, and DIC analyses complement each other, creating a universal method for evaluating the material of concrete with PP particles subjected to fire temperatures.

## Figures and Tables

**Figure 1 polymers-16-01161-f001:**
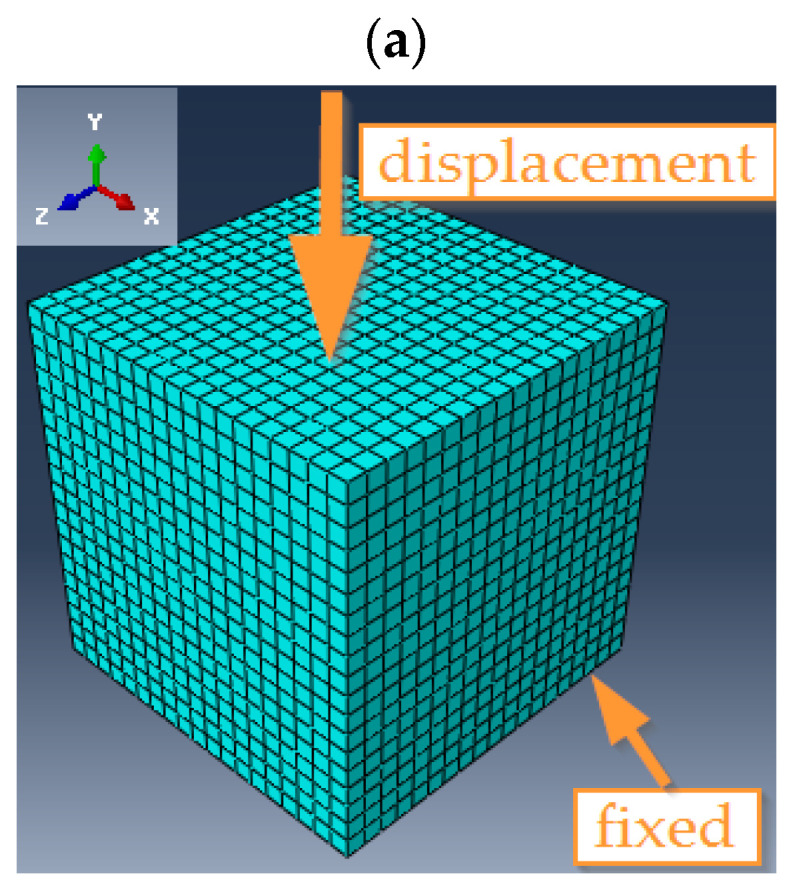
Numerical models developed in Abaqus for (**a**) compression, (**b**) three-point bend specimens.

**Figure 2 polymers-16-01161-f002:**
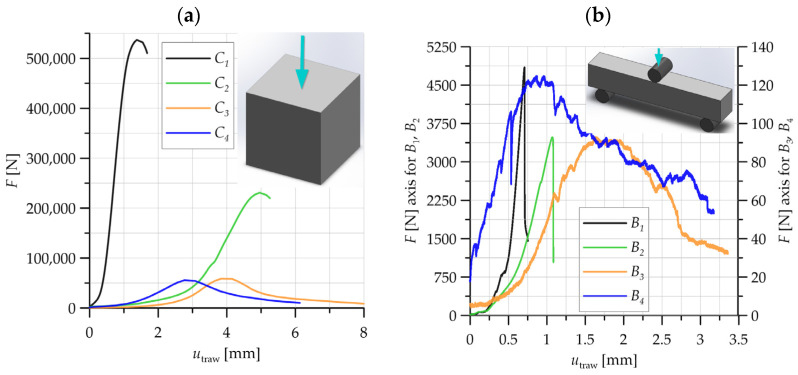
Force–displacement relationships recorded during laboratory tests: (**a**) compression, (**b**) three-point bending of the analyzed concrete.

**Figure 3 polymers-16-01161-f003:**
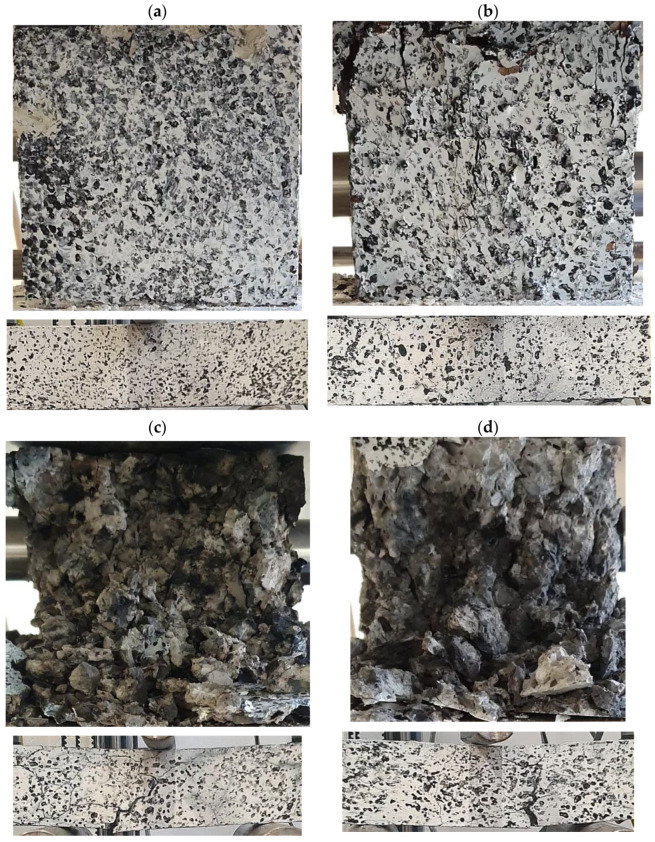
Images of the condition of the specimens after laboratory tests for concrete designated as (**a**) *C*_1_, *B*_1_, (**b**) *C*_2_, *B*_2_, (**c**) *C*_3_, *B*_3_, (**d**) *C*_4_, *B*_4_.

**Figure 4 polymers-16-01161-f004:**
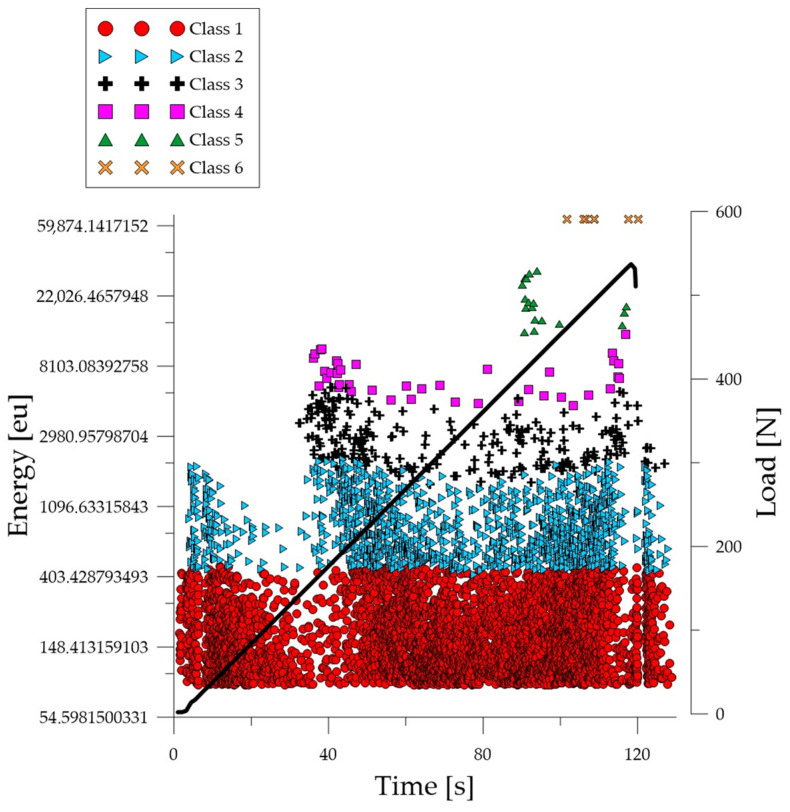
Time course of individual classes of AE signals for the energy parameter for an exemplary specimen from the *C*_1_ series.

**Figure 5 polymers-16-01161-f005:**
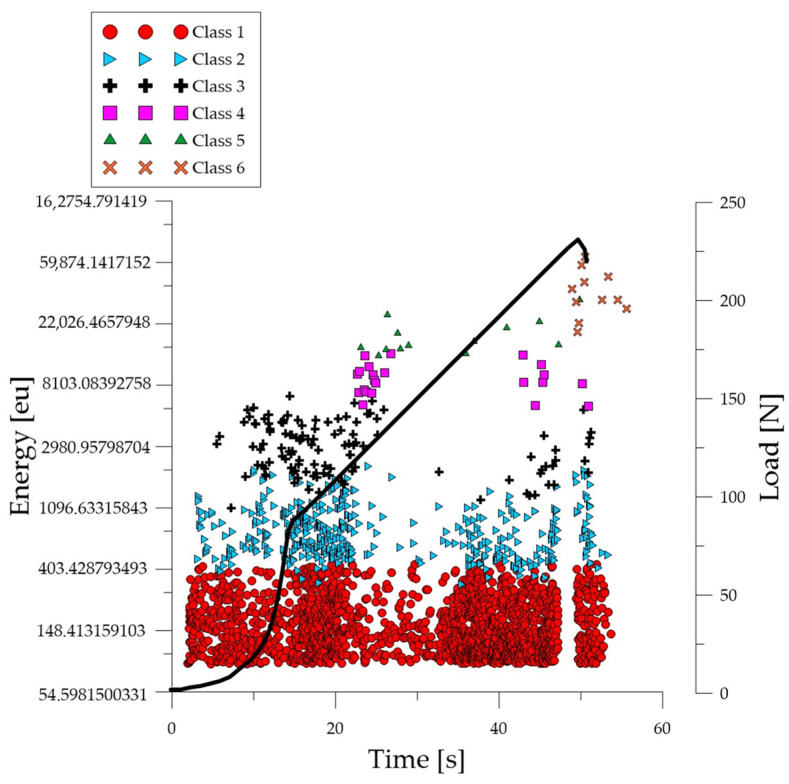
Time course of individual classes of AE signals for the energy parameter for an exemplary specimen from the *C*_2_ series.

**Figure 6 polymers-16-01161-f006:**
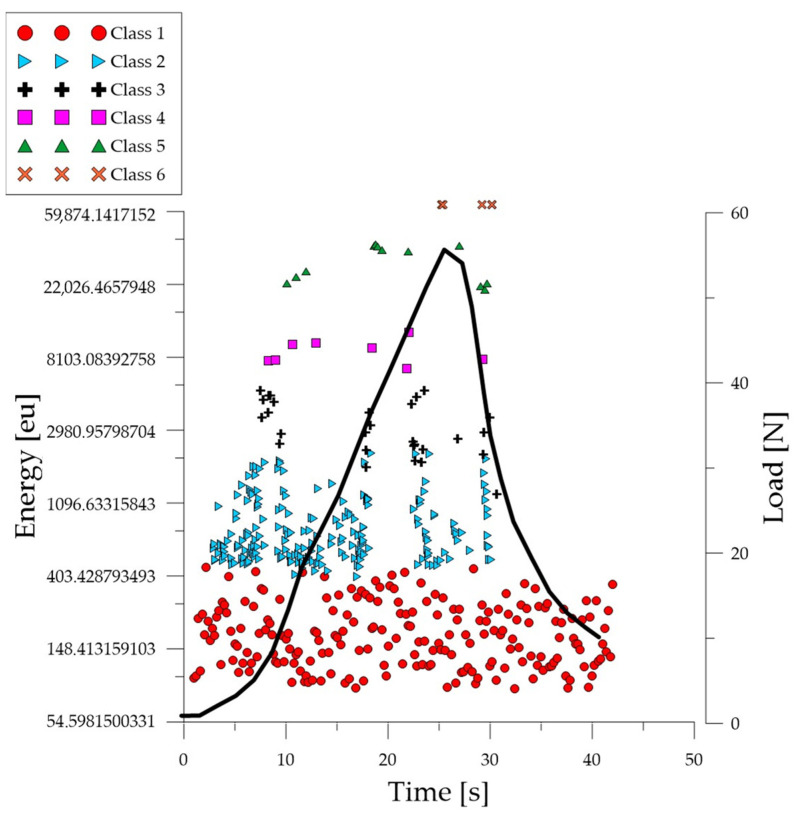
Time course of individual classes of AE signals for the energy parameter for an exemplary specimen from the *C*_3_ series.

**Figure 7 polymers-16-01161-f007:**
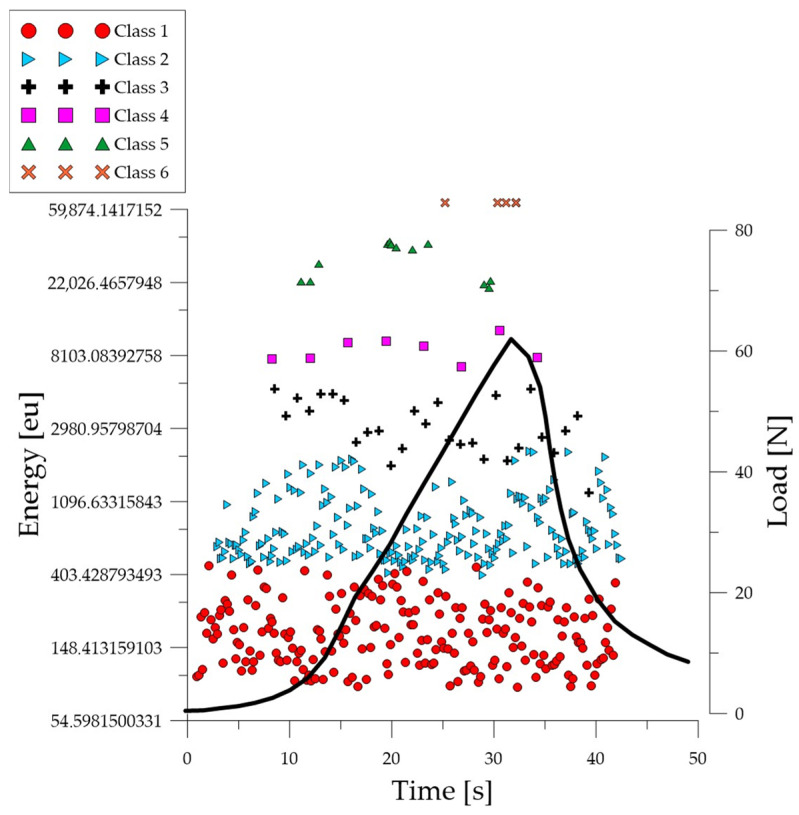
Time course of individual classes of AE signals for the energy parameter for an exemplary specimen from the *C*_4_ series.

**Figure 8 polymers-16-01161-f008:**
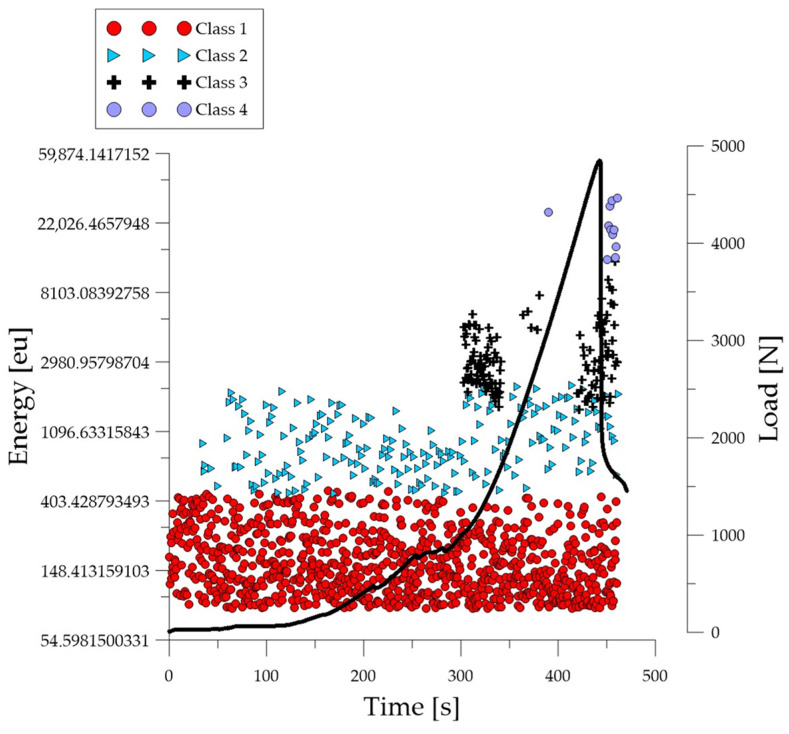
Time course of individual classes of AE signals for the energy parameter for an exemplary specimen from the *B*_1_ series.

**Figure 9 polymers-16-01161-f009:**
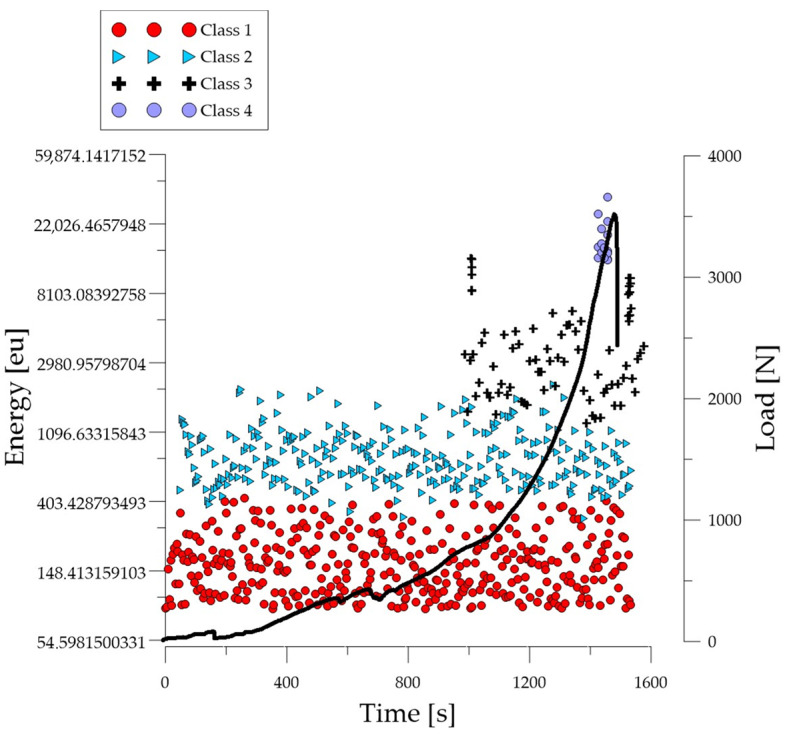
Time course of individual classes of AE signals for the energy parameter for an exemplary specimen from the *B*_2_ series.

**Figure 10 polymers-16-01161-f010:**
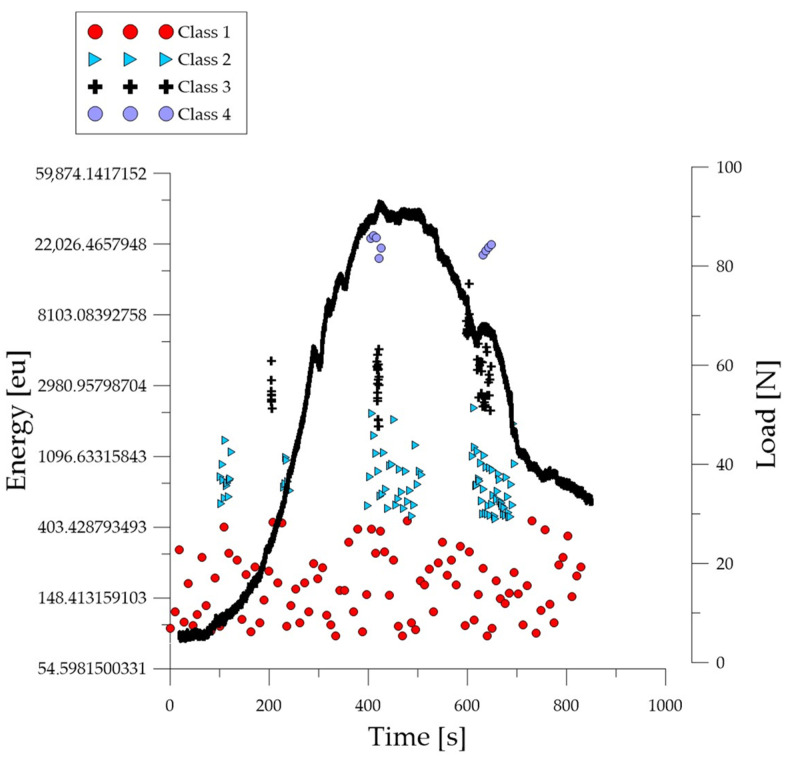
Time course of individual classes of AE signals for the energy parameter for an exemplary specimen from the *B*_3_ series.

**Figure 11 polymers-16-01161-f011:**
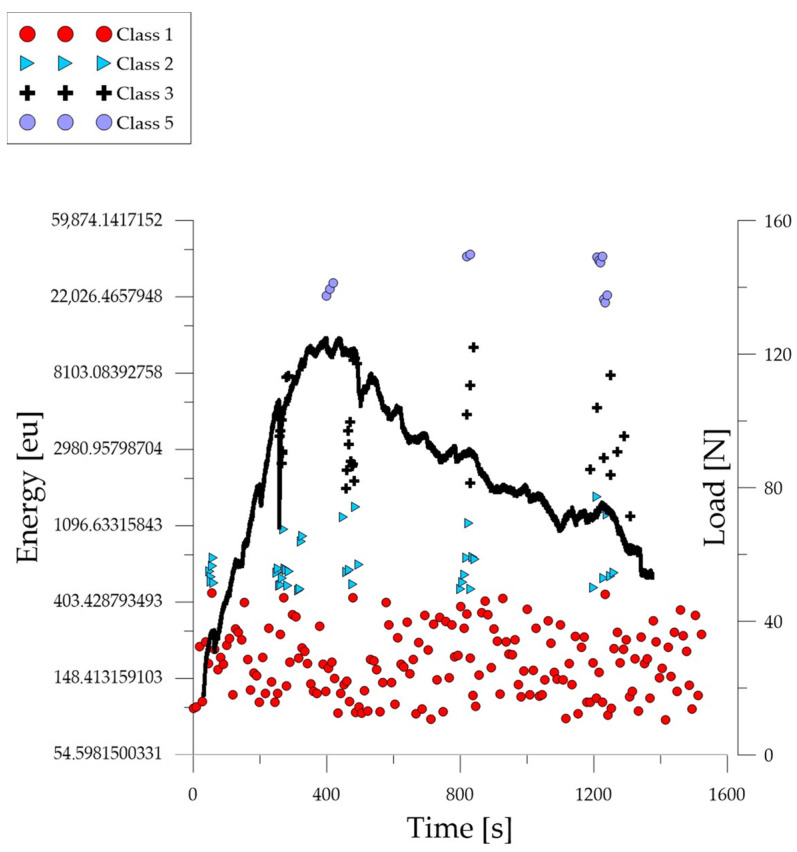
Time course of individual classes of AE signals for the energy parameter for an exemplary specimen from the *B*_4_ series.

**Figure 12 polymers-16-01161-f012:**
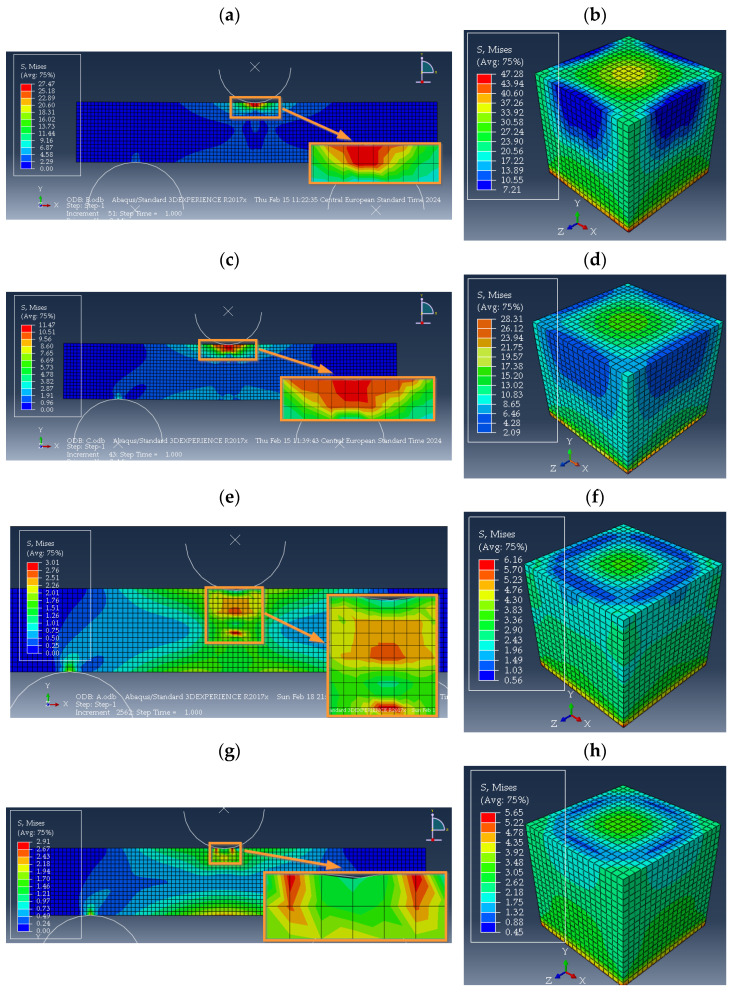
Numerically determined effective stress maps in specimens: three-point bending for materials: (**a**) *B*_1_, (**c**) *B*_2_, (**e**) *B*_3_, (**g**) *B*_4_ and compression for materials: (**b**) *C*_1_, (**d**) *C*_2_, (**f**) *C*_3_, (**h**) *C*_4_.

**Figure 13 polymers-16-01161-f013:**
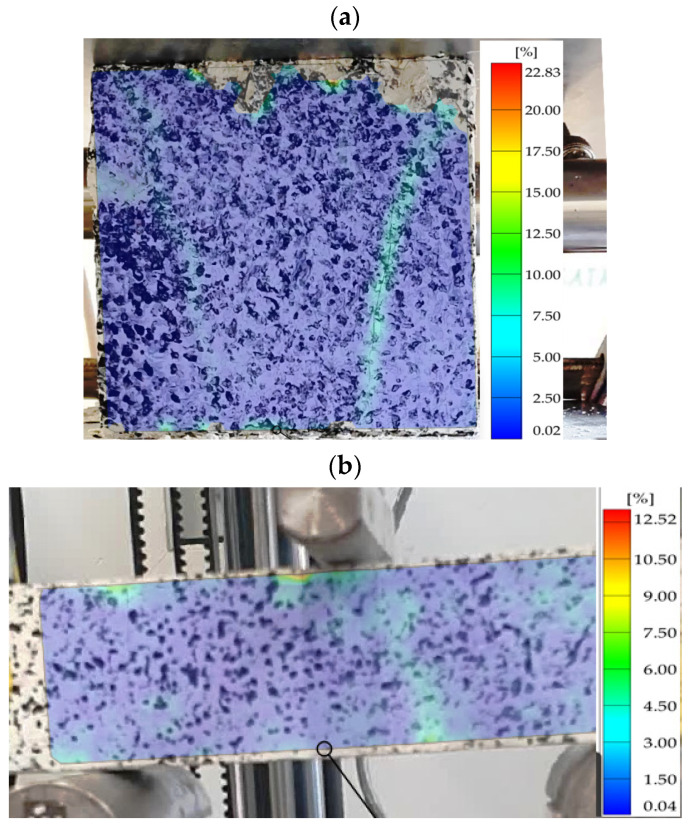
Strain maps obtained on the specimens during the application of digital image correlation (DIC) analysis for concrete designated as (**a**) *C*_1_, (**b**) *B*_1_, (**c**) *C*_2_, (**d**) *B*_2_, (**e**) *C*_3_, (**f**) *B3*, (**g**) *C*_4_, (**h**) *B*_4_.

**Figure 14 polymers-16-01161-f014:**
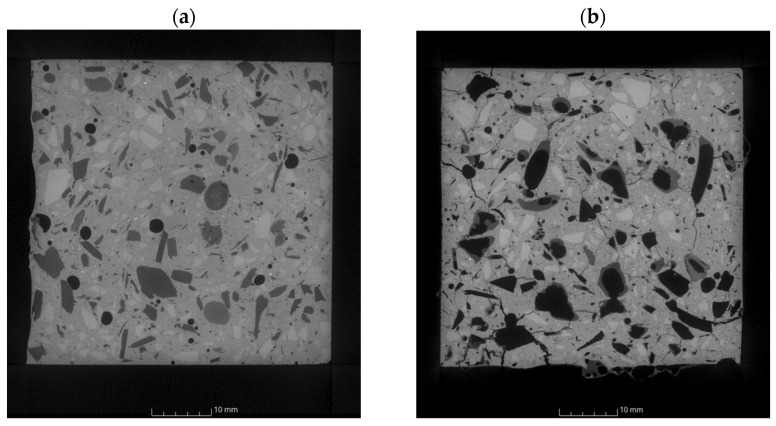
Computed tomography images obtained from the analyzed specimens: (**a**) *B*_1_, (**b**) *B*_2_, (**c**) *B*_3_, (**d**) *B*_4_.

**Figure 15 polymers-16-01161-f015:**
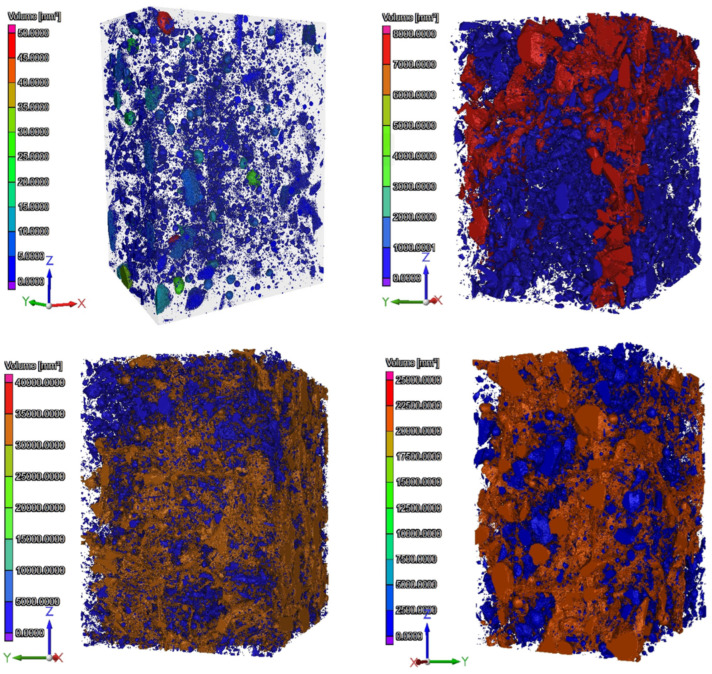
Three-dimensional models of tested specimens with porosity analysis.

**Table 1 polymers-16-01161-t001:** Details of the material model (CDP) used in the numerical simulation program.

Material Characterics/Specimen	*B*_1_, *C*_1_	*B*_2_, *C*_2_	*B*_3_, *C*_3_	*B*_4_, *C*_4_
*E*, GPa	24	16	8	7
*ν*	0.2	0.2	0.2	0.2
Dilation angle	30	30	30	30
Eccentricity	0.1	0.1	0.1	0.1
*f*_b0_/*f*_c0_ (i.e., *σ*_b0_/*σ*_c0_)	1.16	1.16	1.16	1.16
* K *	0.67	0.67	0.67	0.67
Viscosity parameter	0	0	0	0
*σ*_cu_, MPa	23.87	10.08	2.6	2.47
*σ*_tu_, MPa	5.12	3.47	1.44	1.4

**Table 2 polymers-16-01161-t002:** Characteristics of AE signals for individual grouping classes (series C).

Class	1	2	3	4	5	6	7
Signal strength (pV∙s)	5.40 × 10^5^– 2.92 × 10^6^	1.97 × 10^6^– 1.35 × 10^7^	8.17 × 10^7^– 2.03 × 10^8^	3.60 × 10^7^– 8.41 × 10^7^	3.60 × 10^7^– 4.38 × 10^5^	1.20 × 10^8^– 4.26 × 10^8^	5.03 × 10^8^– 8.63 × 10^9^
Max amplitude (V)	88	94	95	96	96	96	97
Max FFT Real (V)	±180	±2000	±20	±250	±8	±180	±2000
Frequency [kHz]	12–87	13–70	24–54	15–58	22–56	3–52	0–35
Max energy (eu)	445	2167	32,473	6812	13,475	65,535	61,535
Max duration (µs)	18,679	57,367	419,761	101,569	169,780	925,479	1 × 10^7^

**Table 3 polymers-16-01161-t003:** Characteristics of AE signals for individual grouping classes (series B).

Class	1	2	3	4
Signal strength (pV∙s)	5.40 × 10^5^– 2.92 × 10^6^	1.97 × 10^6^– 1.35 × 10^7^	8.17 × 10^7^– 2.03 × 10^8^	3.60 × 10^7^– 8.41 × 10^7^
Max Amplitude (V)	88	94	95	96
Max FFT Real (V)	±180	±2000	±20	±250
Frequency [kHz]	12–87	13–70	24–54	15–58
Max Energy (eu)	445	2167	32,473	6812
Max duration (µs)	18,679	57,367	419,761	101,569

**Table 4 polymers-16-01161-t004:** Maximum stress values determined by numerical calculations.

Compression Test				
Material	*C* _1_	*C* _2_	*C* _3_	*C* _4_
Mises stress [MPa]	47.28	28.32	6.26	5.65
*σ*_22_ [MPa]	50.38	30.45	6.69	6.10
Bending test				
Material	*B* _1_	*B* _2_	*B* _3_	*B* _4_
Mises stress [MPa]	27.47	11.47	3.01	2.91
*σ*_22_ [MPa]	12.70	7.78	0.87	0.86

**Table 5 polymers-16-01161-t005:** Comparison of the strain results obtained for the analyzed concrete annealing cases with numerical calculations and the results of DIC analyses.

Compression Test				
Material	*C_1_*	*C* _2_	*C* _3_	*C* _4_
Effective strain—FEM	23.56	57.51	52.12	49.89
Effective strain—DIC	22.83	59.64	50.04	47.84
Differences [%]	3.10	3.70	3.99	4.11
Bending Test				
Material	*B* _1_	*B* _2_	*B* _3_	*B* _4_
Effective strain—FEM	12.72	23.65	42.00	23.89
Effective strain—DIC	12.52	23.07	43.24	24.70
Differences [%]	1.57	2.45	2.95	3.39

## Data Availability

Data are contained within the article.

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
