# Peer review of "Detection of Destructive Processes and Assessment of Deformations in PP-Modified Concrete in an Air-Dry State and Exposed to Fire Temperatures Using the Acoustic Emission Method, Numerical Analysis and Digital Image Correlation"

_polymers, 2024, doi:10.3390/polym16081161_

Round 1
Reviewer 1 Report
Comments and Suggestions for Authors
The manuscript presents a comprehensive investigation into the condition of PP-modified concrete using a combination of experimental and numerical methods, including acoustic emission (AE) testing, Digital Image Correlation (DIC), finite element modeling (FEM), and microstructural analysis. since in my opinion a major revision is required, please reply to the following questions carefully.
Please explain the importance of recording signals such as test time, force applied to the specimen, and displacement of the machine traverse during laboratory tests.
How did the maximum force recorded during compression tests vary with different annealing temperatures of the concrete material?
In the three-point bending tests, what differences were observed in the maximum loading force for specimens subjected to different annealing temperatures?
What were the visible differences in the appearance of specimens after compression and three-point bending tests for different annealing temperatures?
What parameters were considered in the acoustic emission (AE) signals analysis, and why were these specific parameters chosen?
How did the energy parameter of AE signals change over time for specimens from different series (C1, C2, C3, C4)?
What were the four classes into which the acoustic emission signals for bending specimens were divided, and what processes were attributed to each class?
What stress parameters were analyzed in the numerical simulations, and what were their characteristics for concrete with PP particles after exposure to high fire temperatures?
Please explain the observed distribution of cracks in the material cross-section under different annealing temperatures, and how did the presence of PP particles influence crack formation.
In order to provide a broader context and support for their research, the authors are encouraged to incorporate these references into the introduction of the manuscript.
-The influence of basalt fiber on the mechanical performance of concrete-filled steel tube short columns under axial compression
-Property Assessment of High-Performance Concrete Containing Three Types of Fibers
-Utilization of antimony tailings in fiber-reinforced 3D printed concrete: A sustainable approach for construction materials
Comments on the Quality of English LanguageMinor editing of English language required
Author Response
Dear Reviewer,
Thank you for your time and comments on our work. We have included the answers to the review questions in the attached file.

Reviewer 2 Report
Comments and Suggestions for Authors The authors carried out a large amount of research that has practical and scientific significance. However, there are a number of following comments that will improve its quality: 1. It is not recommended to use abbreviations in the title of the article; 2. Information about the relevance of the research must be added to the Abstract section 3. The goal in the article should be adjusted according to the title (here the goal does not mention the Digital Image Correlation method); 4. It is recommended to provide the name of the company that provided concrete samples for research; 5. It would be great to add clarification regarding the selected temperatures and the duration of curing samples at a given temperature. Perhaps there is a standard? 6. L 200. Source 36 cited twice. Correct this; 7. L227. “.. digital image correlation (DIC)...” The abbreviation of this method has already been given earlier. Therefore, here remove either the abbreviation or the full name of the method; 8. It is necessary to explain how the results obtained using CT computed tomography correlate with the results obtained using AE and DIC methods; 9. Figure 3. Please adjust the dimensions of the sample photographs so that they are the same in height; 10. Section 4. "Discussion" in its present form looks like conclusions. It needs to be adjusted or removed.
Author Response

(The authors gave the same response as above.)

Round 2
Reviewer 1 Report
Comments and Suggestions for Authors
Based on the authors' satisfactory response provided, I recommend accepting the manuscript.
Reviewer 2 Report
Comments and Suggestions for Authors
The author has made all the necessary corrections, the manuscript can be accepted in present form